# An Inventory of Good Management Practices for Nutrient Reduction, Recycling and Recovery from Agricultural Runoff in Europe's Northern Periphery and Arctic Region



**Aleksandra Drizo** [1,*] , **Chris Johnston** [2] **and Jón Guðmundsson** [3]

1   International College Science and Sustainability Management Program, Tunghai University, Taichung 407224, Taiwan
2   Agri-Food and Biosciences Institute, Large Park, Hillsborough BT26 6DR, UK; chris.johnston@afbini.gov.uk
3   Faculty of Agriculture and Environmental Sciences Agricultural, University of Iceland Árleyni 22, 112 Reykjavík, Iceland; jong@lbhi.is
*   Correspondence: adrizo@thu.edu.tw; Tel.: +886-4-2359-0121

**Abstract:** The excess loading of nutrients generated by agricultural activities is a leading cause of water quality impairment across the globe. Various management practices have been developed and widely implemented as conservation management strategies to combat water pollution originating from agricultural activities. In the last ten years, there has also been a widespread recognition of the need for nutrient harvesting from wastewaters and resource recovery. In Europe's Northern Periphery and Arctic (NPA) areas, the expertise in water and runoff management is sporadic and needs to be improved. Therefore, the objective of this research was to perform a comprehensive review of the state of the art of Good Agricultural Practices (GAPs) for the NPA region. A set of questionnaires was distributed to project partners combined with a comprehensive literature review of GAPs focusing on those relevant and/or implemented in the NPA region. Twenty-four GAPs were included in the inventory. This review reveals that there is a large level of uncertainty, inconsistency, and a gap in the knowledge regarding the effectiveness of GAPs in nutrient reduction (NRE), their potential for nutrient recycling and recovery (NRR), and their operation and maintenance requirements (OMR) and costs. Although the contribution of GAPs to water quality improvement could not be quantified, this inventory provides a comprehensive and first-of-its-kind guide on available measures and practices to assist regional and local authorities and communities in the NAP region. A recommendation for incorporating and retrofitting phosphorus retaining media (PRMs) in some of the GAPs, and/or the implementation of passive filtration systems and trenches filled with PRMs to intercept surface and subsurface farm flows, would result in the enhancement of both NRE and NRR.

**Keywords:** water quality; agricultural management practices; resource recovery

## 1. Introduction

While agriculture represents an important sector of the economy of Europe's Northern Periphery and Arctic (NPA) areas, its activities impose a significant risk to the susceptible environment through water and land pollution. Moreover, the expertise in water and stormwater runoff management in the NPA region is dispersed and unevenly distributed and needs to be augmented to secure the protection of natural resources while promoting sustainable economic growth. The European Commission's Guide on Best Environmental Management Practice for the Agriculture Sector [1] highlighted that water quality objectives set by the Water Framework Directive (WFD) require new conservation practices, tools and solutions to prepare local and regional authorities and communities for current and future environmental and socio-economic challenges. Given the lack of expertise in the NPA region, interdisciplinary and international collaboration is vital for enabling knowledge and technology transfer and promoting innovation in good management practices.

In 2016, the University of Savonia, Finland, and 23 European partners (Finland, Sweden, Iceland, Faroe Islands, Northern Ireland, Republic of Ireland and Scotland) were awarded 1.7 million euros from the Northern Periphery and Arctic Programme (NPAP) 2014–2020 [2] to investigate the best management practices for agricultural and mineral extraction runoff management (acronym "WaterPro"). The NPAP is an Operational Programme of Interreg covering the NPA region, the North-West Europe Programme, which encourages transnational cooperation to strengthen Northwestern Europe as an economic player, with high levels of innovation, sustainability, and cohesion [3]. This program is supported by the European Regional Development Fund (ERDF) and the corresponding ERDF funding from non-EU partner countries [2].

The overarching goal of the WaterPro project (2016–2019) was to develop eco-efficient instruments and models for surface and belowground water runoff management practices and environmental protection for the sparsely populated region of the NPA. A key outcome of this research was the creation of an Inventory of Good Management Practices for Nutrient Reduction, Recycling and Recovery from Agricultural Runoff. The purpose of this inventory is to serve as a set of guidelines and a tool to enhance the readiness of responsible authorities and local agricultural producers to protect the water quality of coastal and freshwaters, human health, and ecosystems in the NPA region.

This paper provides a comprehensive review of Good Agricultural Practices' efficiencies in nutrient reduction (NRE), their potential for nutrient recycling and recovery (NRR), operation and maintenance requirements (OMR) and costs. It also highlights the gaps in the current knowledge and provides recommendations for future research directions.

### 1.1. Agricultural Management Practices

"Good Agricultural Practices" (GAPs), "Good Management Practices" (GMPs), "Best Environmental Management Practices" (BEMPs) (Europe) or "Best Management Practices" (BMPs) (North America) are commonly defined as "methods and practices designed to reduce or prevent soil and water pollution without affecting farm productivity" [4–6]. They were developed in the 1950s as conservation solutions to mitigate soil erosion and land degradation. They have been implemented as soil remediation practices for two decades prior to their first use to reduce pollution originating from agricultural non-point sources (NPS) and as potential measures to reduce and control the eutrophication of water bodies in 1970s [4].

Despite 5 decades of effort and considerable financial investments in the implementation of GAPs as conservation management strategies for pollution mitigation [1,4–7] the excess loading of nutrients generated by agricultural activities remains the foremost water quality issue in Europe and across the globe [1,8–12]. Consequently, over the past decade there have been a number of large-scale international projects which focused on the development of the guidelines and user manuals of GAPs for the entirety of Europe as well as globally [6,7,12–16]. Today, almost every country of the world has a guide or code of practice for GAPs [12]. A list of key guidelines and reports describing GAPs for agricultural phosphorus (P) and nitrogen (N) pollution mitigation and control in the past 10 years is provided in Table 1.

However, most of the above guidelines focused on the GAPs' descriptions and applications, with very limited information on their treatment efficiency and functionality, costs of implementation, ease of operation and maintenance, potential for nutrient reduction, recycling and/or recovery (N3R) or ability for climate change mitigation. Instead, the guidelines are usually categorized according to the air, soil and water environmental resources degraded by agricultural activities (Table 1). For example, Schoumans et al. [7] provided information regarding 32 GAPs which were grouped as nutrient application management, crop management, soil management, agricultural water management, land use change, land infrastructure and measures in surface waters. The recent guidelines by the European Commission [1] grouped them according to their intended purpose as: (1) soil quality management; (2) nutrient management; (3) soil preparation and crop planning; (4) grass

and grazing management; (5) animal husbandry; (6) manure management; (7) irrigation and (8) crop production products.

**Table 1.** Guidelines and reports describing GAPs for agricultural P and N pollution mitigation and control. Modified from Drizo [12].

| Year | Title | Source |
|---|---|---|
| 2011 | Mitigation options for reducing nutrient emissions from agriculture. A study amongst European member states of Cost action 869. | Schoumans et al. [7] [1] |
| 2011 | An inventory of mitigation methods and guide to their effects on diffuse water pollution, greenhouse gas emissions and ammonia emissions from agriculture. | Newell-Price et al. [13] |
| 2012 | Best management Practices Policy Toolbox Presentation | The United Nations Environment Programme (UNEP) [14] |
| 2014 | Development of the EMAS Sectoral Reference Documents on Best Environmental Management Practice. Learning from frontrunners Promoting best practice. | Schoenberger et al. [6] |
| 2015 | EU Database of Best Practices | Living Water Exchange [15] |
| 2018 | Best environmental management practice for the agriculture sector—crop and animal production | European Commission [1] |

[1] Provides mitigation options and factsheets, as well as a database of mitigation options (http://www.cost869 .alterra.nl (accessed on 16 April 2022).

*1.2. The Effectiveness of Agricultural Management Practices at Reducing Nutrient Losses to Surface Waters*

Many agricultural agencies across Europe and North America have worked with farmers and landowners to implement a variety of agricultural GAPs/BMPs/BEMPs and reduce nutrient and sediment losses to streams and rivers [8–13]. However, determining and documenting the effectiveness of these practices at the field, catchment and watershed scales has been very challenging. Moreover, those studies that succeeded in accessing GAPs' performances revealed that there has been very little reduction in agricultural P pollution and/or improvement in water quality [8–12,17–20].

Barry and Foy [20] revealed that implementing GAPs resulted in significant improvements in water quality in 40 headwater streams in Northern Ireland over a 25-year period. However, they highlighted that many catchments had elevated nutrient concentrations, and there was no improvement in the ecological water quality required by the WFD. Drizo [12] recently reviewed challenges in evaluating the treatment efficiency of GAPs implemented to mitigate agricultural P pollution. She highlighted the extreme complexity of solving the pollution problems which originate from a variety of diffuse sources (e.g., a combination of livestock and cropping systems which result in agricultural surface and subsurface runoff, and their interactions). Therefore, the assessments of GAP treatment efficiencies in those cases are further impeded by the issues of scale and the fact that they are implemented on individual farms, while water quality improvement is evaluated at a larger scale (catchment or watershed) [12].

Mulla et al. [21] investigated factors that affect the assessment of the effectiveness of GAPs in decreasing nutrient losses to surface waters in the USA. They concluded that an assessment at the watershed scale had been impeded due to (1) temporal variability in weather, runoff and drainage, which leads to high nutrient loss variability in daily, monthly, and annual nutrient and sediment exports; (2) lack of scientifically rigorous studies of GAPs' effectiveness at the watershed scale; (3) long lag times which occur as a response to land management changes. The authors estimated that due to the vast amounts of N and P accu-

mulated in soil pools over decades of agricultural production, the response to implemented GAPs can take as many as 5 to 10 years. In addition, potential improvements in stream and river water quality may be concealed by previous accumulation and in-stream sediments and nutrients transport; (4) most conservation programs involve a small percentage of the watershed land area and often exclude the most critical pollution source areas; and (5) due to the lack of long-term field datasets, modeling is often used to project responses to management [22–24]. However, modeling studies have many limitations, including uncertainty in many parameters (e.g., soil hydraulic properties, denitrification, mineralization rates, biological N fixation), incomplete representations of field and watershed processes, and limited data regarding models' calibration and validation.

Randall et al. [16] conducted comprehensive research on the effectiveness of the most commonly used GAPs (e.g., vegetated buffer strips, cover/catch crops, slurry storage, woodland creation, controlled animal trafficking and subsoiling) implemented for the improvement of water quality in temperate farming systems in Europe, Canada, New Zealand and northern states of the United States of America. Their study included 718 articles collected from search engines, peer reviewed articles and gray literature. They found that vegetated buffer strips (including woodland buffers) were the most frequently reported agricultural practice (*n* = 364), followed by cover/catch crop (*n* = 245) and slurry storage (*n* = 93). Most studies were conducted in the northern states of the USA (*n* = 256), with the major focus being on buffer strips. The remaining articles originated from Europe, and most were from the UK (*n* = 80), where cover/catch crops were reported marginally more frequently than buffer strips. The most frequently measured water quality parameter in 718 reviewed articles was N (*n* = 473), followed by P (*n* = 178) and sediment (*n* = 165). Most reported measurements were related to buffer strips (209 studies on N, 136 on P and 128 on sediment), followed by cover/catch crops (203 studies on N, but only 24 on P and 28 on sediment and slurry storage (*n* = 58).

The researchers concluded that (1) studies that measured and described the effectiveness of GAPs (interventions) at catchment scale have been lacking, (2) there has been an absence of studies that implemented controls, pre and post water quality measurements and/or multiple sampling points from both field and rivers, and (3) more research is needed to elucidate seasonal variations in the effectiveness of buffer strips, woodland creation and cover/catch crops. The authors also identified knowledge gaps regarding the performance of buffer strips and highlighted that (4) future research should focus on the assessment of the effectiveness of buffer strips in reducing the leaching of organic forms of N or P and (5) gaining a better understanding of the role and impact of cover/catch crops in reducing organic forms of N and P [16].

The gaps and limitations in the research on GAPs can be attributed to the fact that these practices are typically recommended but not required, and therefore practical implementation is voluntary in nature and offered via various governmental monetary subsidies [12,18]. Moreover, funding for evaluating the efficacy of GAPs treatment has been lacking [12].

### 1.3. The Potential of Agricultural Management Practices for Nutrient Recycling and Recovery (NRR)

With the objective of reducing nutrient pollution in waterbodies, two obvious approaches emerge. One is to prevent or reduce surface runoff to water, and the other is to recapture and recycle any nutrient losses. For the past 30 years, most of the research on nutrient recycling and recovery (NRR) from animal waste streams has been focused on animal manure [12,25–27]. Investigations on the potential for NRR from other on-farm sources started to receive more attention relatively recently, along with a universal recognition of the decline of the world phosphate reserves, in particular those of a high grade [12,28].

Rosemarin et al. [29] recently reviewed a series of systematic reviews and expert opinions on circular solutions for the recovery and reuse of nutrients from agriculture and wastewater effluents. They provided a summary of technologies and practices for nutrient capture and reuse in agricultural applications including contour ploughing, buffer strips,

constructed wetlands, cover crops and anaerobic digestion. The information provided focused on the practice's efficiencies in nutrient retention, while the potential for recovery was only reported for anaerobic digestion, which can achieve N and P recovery rates of over 50% [29].

The authors also highlighted well-known problems associated with the reuse of manure, crop residues, digestates and compost on croplands to improve nutrient reuse efficiency.

Determining the correct quantities of N and P to meet the requirement of the crops is extremely challenging, as for these organic fertilizers, matching N requirements to the crop requirement results in excessive amounts of P being applied to the fields [8–11,18,29].

Drizo [12] recently suggested that incorporating P-retaining materials (PRMs) into GAPs could result in an increase in NRE as well as provide an opportunity for P capture and recovery. She outlined two crucial steps that ought to be made prior to PRMs' use for this purpose. Firstly, plant P availability in the spent filtration material needs to be determined [30,31]. The second step is to perform leachate studies to ensure that there is no leaching from the media that could cause adverse environmental effects in the surrounding environment [12]. Much more research is needed to evaluate the potential and most appropriate ways to recover nutrients from GAPs [12].

### 1.4. The Costs of GAPs Implementation, Operation and Maintenance

Sidemo-Holm et al. [17] highlighted that farmers are not paid for achieving a desired environmental benefit but are instead compensated for their costs in adopting land management measures to protect the environment, and this technique has been criticized as being ineffective. A study conducted by the Organization for Economic Co-operation and Development (OECD) reported that the implementation of GAPs at the local, catchment, regional, national, and international scales requires billions of taxpayers' dollars annually [32]. Financial aid from various governments typically includes (1) agro-environmental payments provided directly to agricultural producers as compensation for a loss of income for adopting sustainable agricultural conservation management practices and (2) disbursements for various forms of technical assistance for GAPs/BMP implementation [18]. Within the EU27, these payments account for 70% of the Common Agricultural Policy budget [33].

In the USA, the implementation of BMPs is funded via the US Department of Agriculture (USDA) Farm Service Agency Conservation Reserve Program (CRP), which is a cost-share and rental payment program. This program's budget provides hundreds of millions of dollars in federal funds annually for the implementation of BMPs [1,34].

Rosemarin et al. [29] reviewed economic tools and measures used to capture and reuse nutrients. They pointed out that specificity and varying external costs make it difficult to draw conclusions regarding the cost–benefits of individual technologies and practices.

Individual practices have different requirements besides the cost of their operation or establishment. They might need special technical skill and other qualifications of the farmer, and thus they are not as easy to implement. Information regarding the OMR is often found in GAPs or other data such as fact sheets.

Various GAPs have been developed and widely implemented for diffuse pollution management; however, they are not always effective. There is a way to improve appropriate focus on specific GAPs for the NPA region to improve on efficiency, efficacy as well as management, maintenance, and operational costs.

The main research questions this paper aims to answer are: (1) which Good Agricultural Practices are most applicable for the management of agricultural nutrient runoff in the NPA region? (2) How effective are they in nutrient reduction (NRE)? (3) What is their potential for nutrient recycling and recovery (NRR), and 4) what are their operation and maintenance requirements (OMR) and costs?

## 2. Methods

To create the GAP Inventory for the NPA Region, a set of questionnaires were prepared and distributed to WaterPro project partners during the summer and fall of 2017. The first

round focused on three key questions aimed at providing specific information regarding the current state of knowledge on GAP use in the NPA Region. These included

(1)    Is there a Code of Good Practice for the prevention of environmental pollution from agricultural activities for the partner country/region?
(2)    Is there any legislation (regulatory requirements) for nutrient (phosphorus, nitrogen or both) removal from agricultural sources (effluents and runoff)?
(3)    What are the current practices recommended in the Code for the management of agricultural nutrient runoff?

Additionally, each project partner was asked to provide a full list of GAPs used in each of the countries/regions, along with any references. This information was obtained from the specialists in the field and responsible regulatory agencies in each partner country/region. Gaps in the knowledge, needs and latest research were discussed during the two project meetings held in Iceland [35,36] and Finland [37] during 2017. The information gathered from the project members is presented in Tables 2–4.

**Table 2.** NPA country/region code of practice for the prevention of environmental pollution from agricultural activities.

| NPA Country/Region | Code of Practice |
| --- | --- |
| Finland | While Finland has a long history of environmental protection and in particular for forest management, e.g., Best Practice Guidelines for Sustainable Forest Management [38,39], most of their documents are in the Finish language, and they do not have a guidelines document specific to Good Agricultural Practices [40]. As an EU member country, the European Commission Guide [1] can be followed for recommendations. |
| Iceland | Good practice guidelines for agriculture and their implementation are compiled and published by the Environment Agency of Iceland (Umhverfisstofnun) in close cooperation with the Farmers Association and Advisory Centre Guidelines supporting farmers in preventing/minimizing pollution from agriculture [41]. |
| Faroe Islands | Government regulation regarding fertilizing with slurry to avoid runoff [42]. |
| Scotland | Government of Scotland published a Code of Good Practice in 2005 [43]. This document provides practical guidance for farmers, contractors and landowners for minimizing the risks of environmental pollution from farming operations. |
| Northern Ireland (NI) | A Code of Good Practice (CGP) for the Prevention of Pollution of Water Air and Soil Guidelines for Northern Ireland (NI) is available at the Department of Agricultural Environment and Rural Affairs website [44] The CGP contains statutory management requirements (SMRs) and good agricultural and environmental conditions (GAECs), and under cross-compliance farmers must follow the practices recommended by these guidelines if they are to claim EU Single Farm Payments or other direct farm subsidies. The SMRs covered under cross-compliance include Nutrients Action Program regulations 2019 [45], Phosphorus (use in Agriculture) Regulations 2015 and Control of Pollution (Silage, Slurry and Agricultural Fuel Oil) Regulations (Northern Ireland) 2003 [46]. However, as with other NPA countries, the CGP does not contain a clear inventory of GAPs for nutrient reduction and recycling and or recovery (N3R). Instead, the CGP guidelines are categorized according to the air, soil and water environmental resources degraded by agricultural practices. |

**Table 2.** *Cont.*

| NPA Country/Region | Code of Practice |
|---|---|
| Republic of Ireland (RI) | European Union (Good Agricultural Practice for Protection of Waters) Regulations 2014, Irish Statue Book S.I. No. 31 published in 2014 [47]. As an EU member country, recommended GAPs are also listed in the European Commission Guide [1]. |

**Table 3.** NPA country/region water legislation for the prevention of environmental pollution from agricultural activities.

| NPA Country/Region | Legislation/Rule | Source |
|---|---|---|
| Finland | <ul><li>Finland's Program for the Protection of the Baltic Sea 2002.</li><li>2005 Action Plan for the Protection of the Baltic Sea and Inland Watercourses.</li><li>River Basin Management Plans 2010–2015 (2009)</li><li>The implementation program of the River Basins Management Plans (RBMP) (2010)</li><li>Government Decree on Limiting Certain Emissions from Agriculture and Horticulture</li></ul> | Nyroos [48] Finlex [49] |
| Iceland | Based on EU Nitrates Directive (regulation 91/676/EC [2]) on water protection against agricultural pollution. EU regulation transposed to Icelandic legislation by regulation 804/1999. | Loftson [35] EUR-LEX European Union Law [50] |
| Faroe Islands | <ul><li>The oldest code of law found in the Faroe Islands regarding livestock (sheep) farming practices is a Royal Decree from 1298 named the Sheep Letter.</li><li>Circular regulating sheep number on each farm to control grazing pressure 1873</li><li>Act about management of sheep farming. Hagalógin 1937 [3].</li><li>2012 Regulation regarding fertilizing with slurry to avoid runoff.</li><li>The constitutional status of the Faroe Islands and foreign relations</li></ul> | Poulsen et al. [51] Faroeislands [52] The Government of Faroe Islands [42] |

**Table 3.** *Cont.*

| NPA Country/Region | Legislation/Rule | Source |
|---|---|---|
| Scotland | Based on EU Nitrates Directive (regulation 91/676/EC) on water protection from agricultural pollution in Nitrate Vulnerable Zones (NVZs).<br><br>• The Private Water Supplies (Scotland) Regulations 1992 set out a maximum admissible nitrate concentration in water of 50 mg/L and are implemented by the local authorities (6A.1)<br>• 6A.2 The Protection of Water Against Agricultural Nitrate Pollution (Scotland) Regulations 1996 transposed into Scots law the requirements of EC Nitrates Directive (91/676/EEC).<br>• 6A.3 Action Program for NVZs (Scotland) Regulations 2003.<br>• Control of Pollution (Silage, Slurry and Agricultural Fuel Oil) (Scotland) Regulations 2003 [53]. | EUR-LEX EU Law [50] UK GovernMent [54] |
| Northern Ireland (NI) | • Nitrates and Phosphorus Regulations 2007–2010<br>• Nitrates Action Program (NAP) 2011–2014 and Phosphorus Regulations [4] | DAERA NI [44] DAERA NI [45,46] |
| Republic of Ireland (RI) | • Department of Agriculture, Food and the Marine (DAFM) handbook sets out the requirements and standards (13 SMRs and the 7 Good Agricultural and Environmental Conditions (GAEC) applicable from 1 January 2015, following the reform of the Common Agricultural Policy) set down in EU legislation (Directives and Regulations) that farmers must comply with. | Government of Ireland [55] |

[2] Council Directive 91/676/EEC of 12 December 1991 concerning the protection of waters against pollution caused by nitrates from agricultural sources [35,50]. [3] Available online in Farois language only, https://heimabeiti.fo/112 (accessed on 30 April 2022). [4] Compliance with the Nitrates Action Program is one of the cross-compliance SMRs. Therefore, farmers claiming the Basic Payment Scheme and other direct payments are required to comply with the NAP Regulations. Measures relating to Phosphorus Regulations are not Cross-Compliance Verifiable Standards. However, adherence to both sets of Regulations is required by law.

**Table 4.** Good Agricultural Practices—nutrient reduction efficiencies (NRE), potential for nutrient recycling and recovery (NRR) and operation and maintenance requirements (OMR) and costs.

| Criteria | Practice |
|---|---|
| \multicolumn | 1. Soil Quality Management |
| | 1.1. Soil Quality Assessment |
| NRE | Current research is limited and non-conclusive. |
| NRR | Current research is limited and non-conclusive. |
| OMR | Management indicators: according to the EC BEMP Guide [1], field soil tests ought to be conducted every 3–5 years for P, K, Mg, pH, OM and bulk density; fields need to be inspected weekly for signs of compaction, surface ponding and erosion. Each farm should have a soil map indicating environmentally appropriate levels of soil P, K, Mg, (index or kg/ha), pH, SNS (kg/ha), and trace elements; soil organic matter balance (+/−) ought to be maintained [1] |
| Cost | The EC BEMP Guide reported that in the UK, soil testing costs approximately EUR 12/field (assuming one soil sample), which includes analysis for P, K, Mg and pH accompanied by fertilizer and lime recommendations. Compaction can be assessed with a penetrometer, at about 82 EUR, which could be shared among neighboring farms. Soil maps can be obtained via the Internet, libraries and academic institutes, usually free of charge. In Finland, the basic soil test (soil type, pH, Ca, K, P, Mg, S, conductivity, cation exchange) costs EUR 15/sample and fertility assessment (microbiological activity, C/N-ratio, organic content) are EUR 60/sample. |
| | 1.2. Conservation Tillage |
| NRE | Conservation tillage (CT) has been widely promoted as a method to reduce sediment and nutrient transport from agricultural fields. It is generally accepted that it improves soil conservation, can provide a reduction in soil sheet erosion and non point source pollution, enhance the retention and storage of soil organic matter and promote improvements in soil fertility [56,57]. However, the effects of CT on sediment and nutrient exports in snowmelt-dominated climates is not well known [56,58,59]. |
| NRR | The surface residue cover of conservation tillage may reduce soil disturbance and decomposition, and increase water retention, soil C and soil N. It has been reported that it can also accumulate P in the surface, thus there is some potential for NRR [59,60]. However, more research is needed to determine the potential of this practice for NRR. |
| OMR | Soil needs to be monitored for compaction, and weed control may be required. Up to seven years of continuous maintenance and management may be necessary before the full benefits of these practices can be realized. |
| Costs | Boyle [58] estimated that application of a conservation tilage can save USD 0.14–0.73 (EUR 0.13–0.70) for every dollar of output produced irrespective of farm size. However, the costs of practice implementation and OMR have not been reported. |
| | 1.2.1. No-till farming: direct drilling |

**Table 4.** *Cont.*

| Criteria | Practice |
|---|---|
| NRE | Studies from Scandinavia showed that during warm, wet winters, high losses of particulate-bound P (PP) in runoff may occur from cultivated clay soils. P losses can also occur as dissolved reactive P (DRP), and may account for 9–93% of the total P lost in runoff [59]. Puustinen et al. [60] found that the loss of PP from no-till soil was 30% lower than losses from ploughed soils (1.13 compared to 3.71 kg ha$^{-1}$), but losses of disolved reactive phosphorus (DRP) increased by 348% under no-till (2.02 instead of 0.58 kg ha$^{-1}$). This has been attributed to the release of DRP from dead weeds following glyphosate application [59] and to P leaching from fertilizers retained near the surface [60]. Overall, there is a lack of consensus in the literature on the effect of no-till practices on nitrate leaching. Any variability appears to depend on soil type, whether catch crops were used before spring-sown crops, and also what the various pathways for water movement in structured soils were [61–63]. |
| NRR | No-till may help improve soil properties such as compaction and infiltration, and consequently help nutrient recycling. However, much more research is needed in order to evaluate the potential of this practice in recycling nutrients. |
| OMR | This practice may require the use of specialized equipment and herbicides in some areas. Leaving out the mechanical disruption of the seed bed via ploughing and tilling can increase the potential for the colonization of weeds and other competitrive pests and as such, other methods to ensure commercial crop production are vital [64]. |
| Costs | This practice reduces the costs of fuel, labor, and equipment. Research in the USA showed that no-till, under Pennsylvania corn production, cut labor by 20% (0.82 h per hectare) for minimum tillage and 54% (2.12 h per hectare) for no-till [58]. |
| 1.3. Agricultural Drainage Management (Soil and Ditch/Water) | |
| NRE | The European Commission [1] lists the following measures as BEMP to mitigate tile drainage pollution impacts: contour ploughing, break slopes, the cultivation of tramlines, the avoidance of compaction, low ground pressure-impact tyres on vehicles and erosion risk planning. However, while all of these recommended measures may improve water infiltration and therefore aid in reducing surface flows, their contribution to minimizing N and P loading has not been quantified. |
| NRR | Current research is limited and non-conclusive. |
| OMR | Sediment removal and periodic mowing of vegetation are necessary costs of maintaining effective drain function [7]. Visual inspections for defining ponding (intervals to be defined by local parameters) [1]. |
| Costs | Limited data. |
| 1.3.1. Controlled Drainage | |
| NRE | According to SERA 17 fact sheet on average, controlled drainage can reduce the loss of total nitrogen and total P by 45 and 35%, respectively [65]. |
| NRR | Current research is limited and non-conclusive. |
| OMR | Sediment removal and periodic mowing of vegetation are necessary for maintaining effective drain function [65]. |
| Costs | The costs are site-specific and depend on the type of control drainage system used [65]. |
| 1.4. Agricultural Tile Drainage | |

**Table 4.** *Cont.*

| Criteria | Practice |
| --- | --- |
| NRE | Current research is non-conclusive. Generally, it is believed that tile drainage often reduces sediment and nutrient export in surface runoff because of the reduction in overland flow from tile drained fields. However, as tile drains convey much of the subsurface flow directly to surface waters, they can also serve as conduits for pollution (nutrients, pathogens, pesticides) transport [12,66–68]. |
| NRR | Not directly from tile drainage. However, Drizo [12] developed a simple passive filtration system which can be placed to collect runoff from the tile drainage or other surface and subsurface flows on farms. She also provided evidence that spent P from the filtration media can be used as a slow-release fertilizer [69]. |
| OMR | A tile drainage system requires proper operation, ongoing inspection and maintenance. These are listed in various fact sheets [7,14,70]. |
| Costs | The costs are site-specific [65]. Insufficient data. |
| | 1.4.1. Phosphorus Removal System #782 |
| NRE | The system was developed as the outcome of a decade (1999–2009) of research by Drizo and co-workers on the use of steel slag aggregates (SSA) for P removal from wastewaters [12]. Depending on the media used, a P removal efficiency of 75% to over 90% has been achieved. The Phosphorus Removal System #782 was accepted by the USDA NRCS as the first interim conservation practice for P removal from surface and subsurface flows on farms in 2013. The Standard recommends that the media should have a P retention capacity of at least 0.50 percent of the weight of the materials, or 4.5 kg P/ton of media [12]. |
| NRR | Bird and Drizo [69] showed that spent P media (SSA) have the potential to act as a slow-release P fertilizer. However, more research is needed to quantify the amount of P than can be recovered from different farm pollution sources. |
| ORM | The system is a user-friendly treatment unit with minimal annual operational and maintenance requirements for the owner. It does not require any mechanical or moving parts, and as a passive filtration system eliminates the need for electrical components. By properly monitoring the system performance, periodic maintenance can be performed at the operator's convenience. The owner should visually inspect filters for signs of scum formation or preferential flows after major precipitation/snowmelt events [12]. |
| Costs | The cost of filters depends on the volumes of wastewater that need to be treated, influent and effluent P concentrations and the availability of the SSA filtration media [12]. Majority of the cost is for media transportation (generally EUR 40/ton). The initial capital costs for larger filters (flows 60–150 $m^3$ $d^{-1}$) can be high. However, the filter has a life span of 30+ years and minimum maintenance fee. In general for base flows of up to 20 $m^3$ $d^{-1}$, the system design cost is USD 7500–9000 (EUR 7150–8500) plus the cost of media and transportation and system construction [12]. |
| | 2. Nutrient Management |
| | 2.1. Field Nutrient Budgeting |

**Table 4.** *Cont.*

| Criteria | Practice |
| --- | --- |
| NRE | Studies from the UK showed evidence that for arable land, a reduction of 5 kg N ha$^{-1}$ leached per year was achieved. For grassland, the reported reductions were 1–5 kg ha$^{-1}$ y$^{-1}$ (dairy) and 2 kg N ha$^{-1}$ y$^{-1}$ (beef). With respect to P, expert analysis estimated a 20% reduction from the fertilizer component [14,70,71]. McCrackin et al. [72] suggested that excess nutrients from animal wastes could be applied to land areas with nutrient deficiences and in doing so improve the agronomics by meeting 54–82% of N reduction targets (28–43 kt N reduction) and 38–64% P reduction targets (4–6.6 kt P reduction) in the Baltic Sea. |
| NRR | This practice does not provide NRR. However, incorporating additional practices into the nutrient management plans such as passive filters can harvest P, which can then be reused as a soil amendment measure instead of traditional chemical fertilizers [12] or the addition of phosphorus immobilizing amendments to soil [7,12]. |
| OMR | The principles of operation are set out in the 4 R's (right rate, source, application method and application timing) in order to provide the appropriate amount of nutrients to the crop where and when needed [1]. |
| Costs | Fertilizer and lime prices have increased considerably over the past decade. The European Comission [1] reported that over the last ten years, the price of a tonne of calcium ammonium nitrate (CAN) fertilizer has increased from approximately EUR 150 to EUR 350. In 2013, a tonne of 20:10:10 N:P:K compound fertilizer was EUR 353/t. To establish a N balance on a UK farm costs in the range of EUR 200–500, depending on the current/future farming system and the extent of advisory/consultancy help. However, these estimations exclude education, promotion and start-up costs [73]. UNECE [74] reported that the costs of establishing a nitrogen budget at national level are in the range of EUR 1000 to 10,000 per year. The cost of increasing N use efficiency through improving management range EUR 1.0 to 2.0 per kg N saved. |
| | 2.2. Crop rotation for efficient nutrient cycles |
| NRE | Crop rotation can improve crop root structure over time, and consequently the chemical, biological, and physical structure of the soil. This will improve the OM and nutrient retention and increase the water-holding capacity of the soil. As crops are removed, nutrients are withdrawn or exported from the system. By examining rotations through time, a farmer can make general estimates of the increase or decrease in potentially available nutrients and change their management accordingly [73]. However, the NRE of this practice has not been quantified. |
| NRR | Leaving the land uncovered for a season helps to regenerate soil and its nutrients which were lost through plant uptake and harvest the previous season [1,7]. However, the potential contribution of the practice to NRR has not been quantified. |
| ORM | In many cases, farmers have more than one rotation sequence on their farm due to field variation and business decisions. |
| Costs | Better use of nutrients creates a more balanced nutrient cycle at the field level and helps farmers to maintain nutrient availability [1,7]. IFOAM [73] suggested that the application of this practice results in lower costs and increased profit margins for the farmers. |
| | 2.3. Precision nutrient application |

**Table 4.** *Cont.*

| Criteria | Practice |
|---|---|
| NRE | The introduction of GPS alone (in autosteer) into farm machinery can increase efficiencies by 5–10% through a reduction in overlaps and gaps in fertilizer spreading [75]. In Finland, farm gate balances on farms around the Baltic Sea showed that nutrient surpluses can be reduced effectively with precise application and farming techniques. A balance of $(+/- 20 \text{ kg N/ha})$ indicates a good status. Precision nutrient application results in reduced fertilizer application and a degree of ammonia abatement; however, its efficiency in reducing P losses needs to be further investigated [39,40]. |
| NRR | Reduced fertilizer application and improved crop yield, in particular regarding N management, indicates that nutrient recycling could also be improved. However, there is a lack of research on the potential contribution of this practice to nutrient recycling and or recovery. |
| OMR | According to experience from Finland, all farms can implement some aspects of precision application practices [39,40]. The European Commission Guidelines [1] recommend that farmers know: 1. what nutrients they are applying (to check nutrient content of manures); 2. the quantity they are applying—application rate (to check flow rate from spreader); 3. when the optimum timing for spreading is—to match crop requirement, when soil moisture allows access and when weather is appropriate; 4. how to spread to ensure maximum nutrient delivery and minimum nutrient loss to the environment via gaseous emissions or surface runoff; and 5. where not to spread manures. The relatively recent integration of global positioning systems (GPS) technology into farm machinery affords farmers the ability to improve the efficiency of nutrient application by largely removing human error and variability in fertilizer spreading by following programmed field boundaries and following tramlines. Furthermore, GPS allows for pre-programmed variable fertilizer application corresponding to soil nutrient status maps or crop canopy variability throughout the field. |
| Costs | GPS units can be purchased as mobile stand alone devices (GBP 300–400 (EUR 350–475)) which can therefore be moved from tractor to tractor, or they can be integrated as part of the purchased farm machinery item which will vary in quality, accuracy and complexity and as such will be more expensive, up to EUR 11,750. In the UK, the integratioon of precision GPS assisted farming can cost the equivalent of GBP 2/ha to GBP 18/ha (EUR 2.35–21.20/ha), while also incorporating field data (aircraft or tractor-mounted radiometry) for assisted real-time fertilizer spreading added a further GBP 7/ha (EUR 8.25/ha). It should be noted that levelling out spatial field nutrient status for N can return up to an extra GBP 65/ha (EUR 76.5/ha) in a year [1]. |
| 3. Soil Preparation and Crop Planting | |
| 3.1. Mitigate tillage impacts | |

**Table 4.** *Cont.*

| Criteria | Practice |
|---|---|
| NRE | The information is non conclusive. For example, Stevens et al. [76] investigated the effects of minimal tillage, contour cultivation and in-field vegetative barriers on soil erosion and P loss in the UK over a 2-year period. Half of the field was cultivated using a minimum tillage approach, while the other half was conventionally ploughed. The difference in these cultivation techniques revealed no significant reduction in runoff, sediment loss or total P loss. The mixed-direction cultivation treatment increased surface runoff and losses of sediment and P. An increase in surface roughness with contour cultivation reduced surface runoff compared to cultivation up- and downslope in both the ploughed and minimum-tillage treatment areas, but this trend was not significant. Sediment and P losses in the contour cultivation treatment followed a similar pattern to surface runoff. |
| NRR | Data are lacking. |
| OMR | Minimum tillage has associated risks of weed infestation. The EC guidelines [1] recommend that this problem can be managed by skilful crop rotation and practices such as stale seedbeds. Additional labor may be required in order to make changes in the field shape and slope to reduce erosion risks. |
| Costs | According to Newell-Price et al. [13], the cost of implementing reduced or no-till operations are based on a contractor being used and the plough retained for occasional use in difficult seasons. The net effect from selling most cultivation equipment and using a contractor is a saving of GBP 40 (EUR 47)/ha. Schulte et al. [77] reported that the application of min-till across Irish cereal production would lead to a total saving of EUR 43.60 million annually, principally from savings in fuel usage of EUR 29.20 /ha. |
| 3.2. Establish cover/catch crops | |
| NRE | According to Justes et al. [78], nitrate leaching can be reduced by 50% if crops are planted on land destined for spring crops. In a study conducted in Ireland, Hooker et al. [79] found that nitrate concentrations and total N load losses were 38% to 70% and 18% and 83% lower, respectively. Similarly, Premrov et al. [80] reported a significant decrease in groundwater nitrate concentration under mustard cover compared to no cover. Berntsen et al. [81] showed that nitrate leaching can be reduced by approximately 25 kg N/ha as an average for spring cereals on sandy and loamy soil. In Finland, it is estimated that winter plant cover can reduce erosion and nutrient leaching by 10–15%. However, in many parts of Europe there are severe issues of post maize harvest erosion and runoff caused by compaction and nitrate leaching, which are exacerbated by the late dates of harvest into the autumn. Therefore, early harvesting may broaden the window for subsequent crop covering and subsequent nutrient loss reduction. |
| NRR | Data are lacking. |
| OMR | Implementing cover/catch crops requires a high level of knowledge from the farmer or advisor. In particular, successful crop production under northern growing conditions requires specific adaptation mechanisms to cope with climatic exceptionalities and handicaps. Soil type, fit with rotation, weeds, plant pathogens, weather patterns, yield, market price and livestock requirements all need to be considered [82,83]. Where cover crops were established as part of the Nitrate-Sensitive Area scheme, it was shown to be preferable (for agronomic reasons) to destroy the crop in January or February (at the latest) [13]. |

**Table 4.** *Cont.*

| Criteria | Practice |
|---|---|
| Costs | The information is not available. One study reported a cost of implementation of EUR 71.20 /ha (including seed and fuel) [83]. |
| | 4. Animal Husbadry |
| | 4.1. Nutrient Budgeting on livestock farms |
| NRE | The DAERA report [46] outlined that by limiting total N fertilization, and calculating required manure N at 60% utilization efficiency, reductions of 70% and 75% are possible in dairy and pig farms, respectively. Kristensen et al. [83] reported that farms in Denmark were able to reduce N surplus and increases in nitrogen use efficiency (NUE) by ca. 30% in a 5 year period and 50% over 10 years. Whole-farm nutrient budgets have been used effectively in the USA. Limited results showed voluntary BMP on concentrated animal feeding operations (e.g., feedlots) was more effective (30–60% reduction in P accumulation) than mandatory nutrient management plans and buffer strips (5–7% reduction in P accumulation) in reducing nutrient surpluses [83]. The EU Nitrogen Expert Panel generated a comprehensive guidance document for assessing nitrogen use efficiency (NUE) at the farm level [84]. |
| NRR | Data are lacking. |
| OMR | The European Commission BEMPs guidelines [1] provides information on three different nutrient management software tools for developing nutrient budgets in the UK. PLANET, MANNER-NPK and ENCASH (http://www.planet4farmers.co.uk/ (accessed on 5 May 2022)). |
| Costs | According to the European Commission BEMPs guidelines [1], the cost of developing nutrient management plans yearly can be in the region of EUR 200–500 per farm, but will provide a net saving in fertilizer cost |
| | 4.2. Dietary reduction of N and P excretion (ruminants and monogastric) |
| NRE | Reducing nutrient inputs and exports from livestock farming can reduce the overal nutrient requirements and recycled pools. Although the data are fairly limited, European Commission guidelines [1] cited research for typical Danish (Northern Europe) pig production facilities. By using 2 different N content feed mixtures and adding synthetic amino acids N excretion per pig could be reduced from 5.3 kg N to 3.9 kg N. This study also suggested that ammonia emissions could also be reduced by 22%. For all pig farming systems, optimized feeding is expected to reduce the overall N excretion in manure by 32%. The guidelines also suggest that optimized feeding (reducing crude protein from 17% to 14% of dry matter) in UK dairy systems could reduce the overall N excretion from the cattle by approximately 48 kg per cow per year [1,85]. Maguire et al. [86] reviewed dietary strategies for reduced P excretion and resulting improved water quality. They stated that reduction in P overfeeding, the use of feed additives to enhance dietary P utilization, and the development of high available phosphorus (HAP) grains are successful measures to decrease fecal P excretion without impairing animal performance. |
| NRR | Data are lacking. Current information is fairly limited. |

**Table 4.** *Cont.*

| Criteria | Practice |
| --- | --- |
| OMR | The EC guidelines document [1] provides operational data for the dietary reduction in N and P excretion. The guide underlines that energy (as metabolizable energy, ME) and protein (crude protein, CP) are the critical nutrients for practical rationing on farm, as these are the most costly nutrients to supply. CP is a simple measurement of the N content of feed (assumed 16% N for budgeting purposes). Recommended CP and ME requirements for livestock are available in farm reference documents and on their websites [84,87]. |
| Costs | Data are limited. One reference [77] estimated that reduced fertilizer N usage rates per kg produced (i.e., improved NUE) could potentially abate 0.080 Mt $CO_2$eq for Ireland, providing a saving of M EUR 28.9. |
| 4.3. Feed management (Reduce runoff from waste forage) | |
| 4.3.1. Silage runoff management | |
| NRE | Data are fairly limited. A few demonstration projects showed that considerable NREs are achievable. For example, a two year project performed in 2010–2012 on a dairy farm in Vermont, USA showed that the implementation of a single trench filled with P-retaining material (P Trap) to a traditional vegetative treatment area (VTA) management practice increased dissolved P reduction from 58% (VTA) to 84%. Total P reduction was 84% [12]. More recently, Sarazen et al. [88] evaluated a novel treatment system consisting of three treatment tanks with a moving-bed biofilm reactor and paired denitrifying woodchip bioreactors. The system's performance was monitored during 16 storm events throughout 2019. The results revealed a 76% cumulative reduction in the TN mass load, a 71% reduction in the nitrite + nitrate-N load, a 26% reduction in the TP mass load, and a 19% reduction in the soluble reactive P load. However, the treatment system also released ammonium-N. |
| NRR | Drizo [12] suggested that incorporating passive P filters ("traps") could recover P for recycling. However, this needs to be demonstrated and assessed at the field scale. |
| OMR | According to the USDA NRCS [89], silage runoff management starts in the silo and as such, harvesting crops at more than 30%dm will greatly reduce the amount of leachate produced. Good agricultural practice such as removing old material, ensuring clear and clean floors and drains, will all contribute to reduced nutrient loss. The low flow collection and separation area needs to be maintained to capture enough of the low flow volume so that the vegetation downstream in the system is healthy with no kill zones [89]. The solid separation screens and settling pools also need to be maintained. The spreader and the vegetated area require maintenance also. They should be checked regularly to be sure that the high flows are moving through the VTA as sheet flow, and are not concentrated in one area. Additional spreaders (gravel trenches) may be needed at intervals along the length of the VTA. |
| Costs | Data are fairly limited; however, an example from Finland itemizes costs of different parts of the infrastructure such as groundwork, pipeline laying and cesspit tank installation, to be in the region of EUR 2.23/$m^2$, with silage effluent managemnet costs around EUR 1.15/$m^3$. In 2011, the cost of VTA implementation was USD 13,170 (EUR 12,550) for a silo 1 acre in size, and the implementation of a low flow gravity runoff diversion cost USD 2050 per acre area, resulting in a total cost of USD 15,220 (EUR 14,500) per acre. The construction is expected to last for 15 years and the cost of materials is expected to be USD 11,040 (EUR 10,500) [89]. |

**Table 4.** *Cont.*

| Criteria | Practice |
|---|---|
| \multicolumn{2}{c}{4.3.2. Passive filters for Phosphorus retention on farms} ||
| NRE | The PhosphoReduc filter system is a"closed loop" gravity-fed passive filtration system for P removal, recovery and re-use as part of a circular economy for such a limited resource with increasing cost and geo-political concerns specifically. Research over the last decade at the University of Vermont, Drizo developed 6 different classes of filters to mitigate nutrient pollution from concentrated agricultural effluents [90–93]. P removal efficiencies of up to 90% have been achieved from dairy effluents [12,90–93]. |
| NRR | Bird and Drizo [69] showed that P harvested from concentrated dairy farm effluent could be recycled from spent P filtration media and reused as a slow-release P soil amendment. However, more research is needed on the methods, costs and efficiencies of P recovery as well as the quantification of the amounts of P that could be recycled and recovered. |
| OMR | Please see 1.1.1, Phosphorus Removal System #782 |
| Costs | Please see 1.1.1, Phosphorus Removal System #782 |
| \multicolumn{2}{c}{5. Manure Management} ||
| \multicolumn{2}{c}{5.1. Physical Manure Treatment (Solids Separation)} ||
| NRE | The reported efficiencies of solids removal via solid–liquid mechanical manure separators vary; however, new advances in equipment and flocculant applications are improving this process [94]. Screwpresses and decanting centrifuge separation can aid in the reduction in P loss to water bodies by removing P from slurry and digestate by 34% (screw press) and from 30% to 93% (centrifuge—however, this is dependent on many factors, such as what the technology is, what the material is, what chemicals are used, the operation of the process, the machine itself, etc.) [95]. Szogi et al. [96] investigated a high-rate solid–liquid separation system combined with flocculant (polyacrylamide) injection to treat swine manure and reported an 89% reduction in total suspended solids, a 72% reduction in organic N, and a 66% reduction in total P. This separation process also efficiently removed heavy metals such as Cu (88%) and Zn (87%). |
| NRR | The number of technologies/methods applied on farms at the full scale are very limited. Vanotti et al. patented a process to recover phosphate from liquid swine manure using polymers [97]. Screwpresses are becoming more commonplace on large livestock units where nutrient pressures encourage nutrient export from the farm [96–98]. |
| OMR | The gravity separation system involves the use of settling basins where solids settle to the bottom and the liquid portion remains at the top and is pumped out to a separate tank for storage or application. Mechanical separation systems use some form of mechanical process to separate liquids from solids (screwpress, centrifuge, screens). All such systems require some level of supervision, labor and maintenance [94–96]. |

**Table 4.** *Cont.*

| Criteria | Practice |
|---|---|
| Costs | The costs involved in separating solids from liquid manure include the cost of the system, construction and/or installation, energy and labor to operate the system, and system maintenance and repairs. Liu [98] reported that mechanical screwpress separators range from USD 10,000 (EUR 9530) to over USD 50,000 (EUR 47,640), with centrifuges in the region of EUR 58,250–290,000, plus the cost of installation, pumps, sumps and channels. Their throughput capacities (the amount of liquid manure and wastewater processed) vary from 0.4 $m^3$ to 2.4 $m^3$ per minute. |
| | *5.2. Appropriate slurry processing and storage systems* |
| NRE | The European Comission document on Best Available Techniques (BAT) for the Intensive Rearing of Poultry and Pigs [99] provides a comprehensive list of the BATs for slurry processing. The maximum duration of slurry storage depends on the capacity of slurry stores in relation to slurry generation (animal numbers). In Europe, the Nitrates directive requires sufficient storage for the winter closed period to ensure no land application of nutrients [1]. Cuttle et al. [71] reported that increasing slurry storage capacity from an average of three to six months under a cool, temperate, wet climate (UK) resulted in: <br><br> • 25% reduction in slurry P losses to water <br> • For arable land, a 10–20 kg N/ha (20–40%) reduction in annual N leaching via optimized application timing, or a 15–30 kg N/ha (30–60%) reduction if fertilizer application rates are reduced accordingly; <br> • For grassland, a 2–5 kg N/ha reduction in N leaching for dairy farms, and 1 kg N/ha reduction for beef farms. |
| NRR | Data are limited. More research is needed. |
| OMR | According to the EC Guidelines [1], best practice is to install tall (>3 m) slurry storage tanks with a comparatively small exposed surface area (new stores), and to cover slurry with some form of fixed or temporary cover (retro-fit existing stores) to reduce gaseous emissions (NH3, GHGs). Newell-Price et al. [13] recommended several measures to minimize emissions to the environment: (1) increase the capacity of farm slurry (manure) stores to improve timing of slurry applications, (2) adopt batch storage of slurry (slurry should be stored in batches for at least 90 days before land spreading; fresh slurry should not be added to the existing storage during this storage period), (3) install covers on slurry stores and (4) allow cattle slurry stores to develop a natural crust (e.g., retain a surface crust on stores, composed of fiber and bedding material present in cattle slurry, for as long as possible). |
| Costs | Klimont and Winiwarter [100,101] developed a model to estimate storage investment costs for different storage scales. The EC BAT document [99] reported necessary investments and annual costs for four different scales of storage capacities (e.g., 500, 1000, 3000 and 5000 $m^3$). The investments for a 500 $m^3$ storage unit were EUR 100/$m^2$ for a tent roof, EUR 39.5/$m^2$ for floating bricks and EUR 10.2/$m^2$ for light bulk materials, while for a 5000 $m^3$ storage unit they were EUR 46, 39.5 and 7.6/$m^2$, respectively. |
| | *5.3. Appropriate solid manure storage* |

**Table 4.** *Cont.*

| Criteria | Practice |
|---|---|
| NRE | The quantities of manure generated from an intensive livestock agricultural unit can often exceed local crop requirements and areas available for application, posing considerable challenges for environmentally sound nutrient management [102–104]. This is particularly pertinent in temperate regions of the world, where there is a winter ban on the spreading of organic nutrient for up to 6 months (December 15th to April 1st). In addition, in many areas, manure is stored in open pits that can give rise to significant P pollution as a result of rainfall [102–104]. The NRE of solid manure storage is generally small [104]. |
| NRR | None. |
| OMR | The EC guidelines [1] suggest that separation of animal excreta prior to storage is best practice for farms with liquid slurry systems. Farm owners should compost or batch store the solid fractions arising from all manure management systems, and especially farm yard manure and poultry litter. As a general recommendation, the manure storage facility must be located in a well-drained area and surface water should not enter it. In addition, an appropriate effective buffer strip must exist between the manure storage facility and the watercourse [1]. Siting manure heaps away from drains and water courses reduces the risk that preferential flow of effluent through the soil might transport N, P and fecal indicator organisms (FIOs) to field drains. |
| Costs | Handling manure has many costs, including equipment purchasing, operation, maintenance, field manure application, and potential liability costs if there is a spill. Additional costs may be incurred where the land base is limited and additional land must be rented, or in situations where manure agreements must be established. The Eurostats [105] provide thorough information on the manure storage statistics. Manure value and economics is also provided by LPELC [103]. |
| 5.4. Slurry application by injection and manure incorporation | |
| NRE | It has been suggested that slurry application via injection can provide the greatest level of nutrient loss reduction to both atmospheric and surface runoff pathways (including both dissolved and sediment bound nutrients), as well as odor reduction, due to limited quantities of material left on the soil surface, limited soil disruption, and immediate soil closure [1,103,104]. Nutrient loss reductions occur primarily due to reduced opportunity for ammonia-N volatilization and in some cases lower dissolved P and N losses in surface runoff. Nutrient loss reductions may vary with weather and timing between application and soil mixing, degree of soil mixing, and percent soil surface disturbance [106,107]. The EC Guidelines [1] reported NH3 abatement efficiencies of up to 90% with closed-slot deep injection, and 70% for open slot shallow injection. |
| NRR | Data are lacking. |

**Table 4.** *Cont.*

| Criteria | Practice |
|---|---|
| OMR | There are two ways to inject slurry: i) open slot and ii) closed slot. The first one is applied for use in grassland, while the second one is applied either shallow (5–10 cm depth) or deep (15–20 cm). The EC guidelines highlight that the use of deep injection is more limited due to the fact that mechanical damage may decrease the herbage yields on grassland. In addition, there is a considerable risk of N losses as $N_2O$ and $NO_3$. Other potential limitations include the soil depth, soil and clay content, moisture of soil [1]. The recommendations for this particular practice are:<br><br>• Shallow injection application of slurries<br>• Incorporation of manures within one hour of spreading.<br><br>In addition, it should be combined with crop nutrient managemnet (2.1) and precision nutrient applications (2.3). |
| Costs | Injection equipment (tank and injector) represents a high initial cost. In the USA, the cost can exceed USD 100,000 (EUR 95,300) [106]. The Cornell University Extension recommends the following factors to be considered when investing in equipment: (1) the size of and/or the number of animals in the operation, (2) the number of hours the equipment will be used in the field, and (3) the need for nurse trucks and draglines, including equipment, accessories, fuel, labor and operator costs [107]. According to the EC BAT reference document [99], the application of slurry by injection is slower and requires higher tractor costs per unit of slurry spread. In addition, machinery repair costs are higher for band spreaders, due to higher soil/machine contact and more moving parts. |
| | 5.5. Chemical Amendments |
| | 5.5.1. Treating Poultry Litter with Aluminum Sulfate (Alum) |
| NRE | Madrid et al. [108] highlighted that almost no data are available under farm conditions in Europe. Several studies conducted in the US indicated that alum application to poultry litter reduced P in runoff by 87% from small plots and by 75% from small watersheds. Alum reduced ammonia emissions by 70%, which also resulted in a higher nitrogen content of the litter [109–111]. However, data are limited and more research is needed. |
| NRR | No. |
| OMR | Trials conducted in the US suggest that alum should be applied to poultry litter at a rate equivalent to 5–10% by weight (alum/manure) [110]. For typical broiler operations growing six week old birds, this is equivalent to adding 0.045–0.090 kg of alum per bird or 1–2 tons of alum per house per flock for about 20,000 birds in each house. The reduction in ammonia emissions is due to the acid produced when alum is added to the litter [109]. |
| Costs | Limited research conducted in the USA suggests that the use of this practice could result in a yearly economic return of USD 308 (EUR 290) for the grower and USD 632 (EUR 600) for the integrator (company), a combined return of USD 940 (EUR 890) [110,111]. |
| | 5.5.2. Phosphorus Immobilizing Amendments to Soil |

**Table 4.** *Cont.*

| Criteria | Practice |
|---|---|
| NRE | This practice is included in the Cost869 list of practices [7]; however, it is excluded from the EC guidelines [1]. The research on the use of industrial by-products and natural materials as P retaining materials (PRM) was pioneered by several researchers in Europe in the early 1990s [12]. In the US, lime and gypsum have been used for several decades [112]. Bryant and co-researchers developed gypsum "curtains" (gypsum-filled ditches) to adsorb soluble P from the runoff. This work suggested that P runoff could be reduced by 50% and continue doing so for up to 10 years. [113]. Chardon and Dorioz [114] developed phosphorus immobilizing amendments to soil as one of the BMPs to reduce P. Aluminium-based substances ($Al(SO)_4$, $Al(OH)_3$ and $AlCl_3$) at an application rate 10 g Al/m$^2$ achieved a 22–90% reduction in total phosphorus (TP) in spring simulations in Finland. Fly ash also achieved 22% efficiency in reducing P runoff [40]. Uusitalo et al. [115] investigated the effects of gypsum on the transfer of P and other nutrients through clay soil monoliths in Finland. The results from this study show that gypsum-amended soils exhibited substantial decreases in turbidity (45%), particulate phosphorus (70%), dissolved reactive phosphorus (50%) and dissolved organic carbon (35%). The authors concluded that gypsum amendments could have the potential for slowing P loss from agricultural areas. However, more research is needed to identify the most suitable materials, quantities, life span and potential for P recovery/reuse as a sustainable fertilzier [12]. |
| NRR | It is considered that most of the investigated industrial by-products and natural materials used for P retention will have the potential for P recovery for reuse. However, further research is needed to determine the best methods and indeed the quantities of P which can be recovered and then how good a fertilizer it is with respect to the plant bioaviliability of P and its effectiveness as a soil conditioner [12]. |
| OMR | According to Chardon and Dorioz [114], no specific skills or technical equipment is needed. However, this needs to be revised as the use of basic farm equipment (e.g., backhoe loader) is necessary in order to place (and later excavate) PRM in the field. Additionally, visual inspection of the material after strong rain events or snowmelt is necessary to check for clogging. Finnish experience is such that PRMs could be applied on fields during autumn prior to snowfall to prevent P runoff during springtime when frost and snow melt occur [40]. |
| Costs | Cost will depend on the price of the material, the quantities needed and its transportation. Other costs will include labor for soil excavation and PRMs frequency of placement in the field). In Europe, transportation costs can be in the order of EUR 50/metric tonne [12]. |
| 6. Nature Based Systems for Diffuse (Nonpoint) Pollution Sources | |
| 6.1. Vegetative Buffer Strips (VBS) | |

**Table 4.** *Cont.*

| Criteria | Practice |
|---|---|
| NRE | Vegetative buffer strips (VBS), also known as filter strips, biofiltration blocks, buffer strips, and buffer zones, have long been accepted as the most common agricultural practice/mitigation measure for nutrient pollution prevention from diffuse/nonpoint pollution sources across the globe. In EU countries, they are mandatory practice under the Common Agricultural Policy [116,117]. However, their NRE is reportedly highly variable, ranging from below zero up to almost 100%, depending on many factors such as plantation width, vegetation (plant species used), nutrients considered, input load, climate, local hydrogeological conditions, and the time period after installation (crop maturity and establishment) [16,116,117]. Richardson et al. [116] and Georgakakos [118] reviewed the history and performance of fixed-width buffer strips and concluded that despite billions of dollars in investment and 30 years of promotion and implementation on agricultural land worldwide, there has been very little evidence of their efficiency, in particular in P reduction. |
| NRR | None—however, VBS could be retrofitted with PRMs, in which case they could provide P recycling/recovery. |
| OMR | VBS should be inspected after heavy rain/runoff events and checked for debris/litter and sediment accumulation. Depending on the vegetation, harvesting is also required to ensure continued crop P requirement via off-take and thus avoid a build-up of P in the soil [117,118]. |
| Costs | Establishing VBSs requires (1) investments in terms of seeds, plants, soil excavation equipment and labor for construction and planting; (2) assistance of an extension expert may be required to adapt the design to local soil and site conditions. There is not enough information on the costs of implementation and maintenance, as these will depend on many factors such as location, soil type, difficulty of excavation, type of vegetation used, among other factors. According to data from the US, the implementation costs can range from USD 32–74,000 ha$^{-1}$ (EUR 30.5–70,500 ha$^{-1}$) of filter strip. The typical maintenance costs reported for the US are USD 865 (EUR 825) ha$^{-1}$ y$^{-1}$. However, this cost is highly variable and depends on the extent and frequency of maintenance needs [16,116]. |
| | 6.2. Constructed Wetlands |
| NRE | Constructed wetland (CW) NRE from agricultural sources has been poor regardless of the complexity of the design used, especially in cold climates [119–124]. Knight et al. [124] compiled the Livestock Wastewater Treatment Database for North America containing the treatment performance of 38 CW systems. They reported that average TP reductions were highly variable and averaged only 42% for livestock management including cattle feeding, dairy, poultry and swine. Moreover, there is a potential for CWs to become saturated and then a source of nutrient over time if not managed correctly. Kadlec [125] reviewed large CWs for P control which included 66 systems with a median size of 210 ha (2,100,000 m$^2$). He pointed out that higher P reduction (71% on average) was achieved due to a low median hydraulic loading (2.55 cm day$^{-1}$), and that the amount of P stored was just 0.77 g P m$^{-2}$ year$^{-1}$. Nitrogen removal can be enhanced by using artificial aeration [121,122,126]. A review of CWs from Finland showed that the CW's treatment efficiency is highly dependent on the wetland's relative size compared to the upstream catchment area, and on the amount of agricultural land in the upstream catchment [127]. Instead of constructing new wetlands, the restoration of degraded wetlands can serve the same purpose [126]. |

**Table 4.** *Cont.*

| Criteria | Practice |
|---|---|
| NRR | CWs do not provide nutrient recycling/recovery. However, they can be retrofitted with PRMs, in which case they would be able to provide P recycling/recovery [12]. |
| OMR | Operational costs include water quality testing, water level adjustment, weed control, flow distribution and level adjustment sumps. OMR costs can range from EUR 400 per year for surface flow systems to EUR 2000 per year for subsurface flow systems. |
| Costs | The capital costs of CWs depend on a variety of factors including retention time, treatment goals, depth of media, type of pre-treatment and material importation costs. Generally the costs include land, excavation, liners, gravel (subsurface flow systems), plants, distribution and control structures and fencing. In general, the median cost of surface and subsurface flow wetlands is EUR 41,900 per hectare and EUR 340,000 per hectare, respectively [128]. |

Concomitantly, an extensive literature review of GAPs was conducted focusing on those relevant and implemented in the NPA region. The main criteria for inclusion in the NPA GAPs inventory were:

(1) the purpose (e.g., whether their primary goal is to achieve nutrient reduction, treatment efficiency and functionality)

(2) the ability for nutrient recycling and/or recovery.

Some practices are directly designed to capture nutrients before they are released from soil, e.g., catch crops, while others aim at recovering nutrients from the runoff, e.g., constructed wetlands and willow biofiltration blocks and buffer zones. For each of the practices, we reviewed the literature and commented on the general potential of each practice when applicable. Furthermore, we also reviewed information on:

(3) the operation and maintenance requirements and costs of implementation.

## 3. Results

The results from the questionnaires, discussions and workshops are summarized in Tables 2 and 3.

As four of the NPA partners are members of the European Union, the inventory of practices for the region has been structured according to the categorization recommended by the European Commission BEMPs Guidelines [1]. The 6 main categories include (1) soil quality management, (2) nutrient management, (3) soil preparation and crop management, (4) animal husbandry, (5) manure management and (6) nature-based systems for diffuse (nonpoint) pollution sources. We reviewed the state of the art on nutrient reduction efficiencies (NRE), potential for nutrient recycling/recovery (NRR), operation and maintenance requirements (OMR) and costs for 24 GPAs recommended for use in the NPA region (Table 4).

## 4. Discussion and Future Research Directions

The review of the selected 24 GAPs which are or could be applied in the NAP region (Table 4) highlights that there is a large level of uncertainty, inconsistency, and a gap in the knowledge regarding their effectiveness in nutrient reduction (NRE), their potential for nutrient recycling and recovery (NRR), and their operation and maintenance requirements and costs. These results are consistent with the previous findings reported by Drizo [12], who conducted a comprehensive review of agricultural management practices (AMPs) for P reduction, and discussed methods and challenges for evaluating their cost effectiveness.

There has been a strong focus on investigating performance and cost effectiveness of GAPs over the last decade and longer, and there are still many unknowns due in large part

to the vast variability of both the type of GAP and site and conditions for implementation. This was also clearly illustrated by the Land and Policy Journal 2010, which included 12 scientific papers in a Special Issue on soil and water conservation measures in Europe (Volume 27, issue 1). These publications further discussed options and methods by which the performance and cost effectiveness of GAPs could be determined, ultimately leading to more widespread adoption and installation.

In 2012, the *Journal of Environmental Quality* (Issue 2) published 14 scientific papers describing findings from the five-year long research study conducted by the European Cooperation in Science and Technology (e-COST) program which investigated the suitability and cost-effectiveness of different options for reducing nutrient loss to surface and groundwaters at the river basin scale [7]. The following year, 150 delegates participated in the 7th International Phosphorus Workshop (IPW7) held in Sweden, focused on the management of agricultural P to minimize impacts on water quality. These discussions were summarized in a series of papers published in a Special Issue of AMBIO journal in 2015. All of the above studies acknowledged the lack of data and gaps in knowledge regarding the GAPs' performance, OMR and cost effectiveness. Nevertheless, the EU spent over EUR 41 billion per year in direct payments to farmers during the 2014–2020 period alone to support implementation of GAPs to protect water quality [12,129].

Drizo [12] highlighted challenges and complexity in the evaluation of GAPs treatment performances, and the fact that field assessment of agricultural management practices requires the purchase, installation and operation of the advanced monitoring equipment, samples collection and analyses. The automatic flow sampling equipment is very costly. Sequential portable samplers, which are the most frequently used, cost ~USD 6000 a piece [130]. The evaluation of the GAPs' performance in pollutant mass reduction requires the installation of a minimum of two pieces (at the inflow and outflow before and after GAP). For this reason, GAPs' performances are generally evaluated based on grab sampling only (following storm events). While grab sampling enables data collection on pollutant concentrations, it does not provide any information on the actual temporal concentrations or pollutant mass loading or achievable reductions by the GAP which has been evaluated.

The fact that GAPs' NREs remain unknown makes it even more difficult to elucidate their potential for NRR. Moreover, to date, most of the research on nutrient recovery and recycling has been focused on municipal sewage effluents at wastewater treatment facilities (MWWTF), with very limited research on animal manure and other agricultural pollution sources [12,123]. Drizo [12] suggested that some of the reasons for the lack of research on NRR from agriculture may be due to the fact that the costs in the nutrient recovery processes on farms cannot be recovered via the same mechanisms used for MWWTP upgrades and installations, e.g., through water tariffs, or a mix of tariffs, transfers, and taxes, because such a funding mechanism does not exist for agricultural wastewater sources. Therefore, it is much harder to sell and/or ensure return on investment if attempting to promote and offer P and/or N recovery technologies in this market, as funding sources would have to come directly from farmers, e.g., private sources. Additionally, the cost of nutrients recovered from agricultural operations is much higher compared to mineral fertilizers, and there are no economic incentives for farmers to invest in recovery processes. This situation creates a considerable gap in research and development of new processes and technologies for NRR.

During the WaterPro project (2016–2019), an experimental trial was established at the Agri-Food and Bioscience Institute (AFBI) Research Farm near Hillsborough, Co. Down in N. Ireland. Here, willow biofiltration blocks were investigated for their effect on runoff which was directed to separate v-notch weirs with flow-triggered and proportional monitoring of land drainage water. Over the three years of data collection, there was evidence that, by virtue of willow's high evapotranspiration, its effect on drying up the soil and reducing the soil moisture content and by improving the hydraulic conductivity of the soil does lead to a reduction in total hydraulic volume and phosphorus runoff into the collection trough; a proxy for the receiving water environment [131–133].

## 5. Conclusions

In this paper, we provide the state of the art on the 24 GAPs used in the NAP region. There is a current lack and inconsistency of data as well as a knowledge gap in the actual nutrient reduction efficiencies (NRE), potential for nutrient recycling/recovery (NRR), and operation and maintenance requirements (OMR), and therefore costs cannot be accurately quantified. However, this inventory provides a comprehensive and first-of-its-kind guide on available measures and practices to assist regional and local authorities and communities in the NAP region. Therefore, this review paper could be used as a platform to revise the implementation of GAPs and agricultural support payments to make them more goal-oriented and linked to performances and achievements rather than activities.

As investigations of PRMs have advanced considerably over the past 25 years, incorporating PRMs in some of the GAPs (e.g., 5.5.1. Phosphorus Immobilizing Amendments to Soil, 6.1. Vegetated Buffers Strips (VBS) and 6.2. Constructed Wetlands) could increase their potential for P recycling/recovery. Moreover, the implementation of passive filtration systems and trenches to intercept surface and subsurface farm flows (e.g., 1.4. Agricultural Tile Drainage, 2.1. Field Nutrient Budgeting, 4.3.1. Silage Runoff Management, and 4.3.2. Passive Filters for Phosphorus Retention on Farms) would result in enhanced NRE and NRR. Trials on SRC willow buffer zones being conducted by the Agri-Food and Bioscience Institute (AFBI) in N. Ireland are proving that not only does the intervention reduce the outflow of P pollution, but their management also removes P with a strongly positive effect on agricultural greenhouse gas emissions and energy production.

**Author Contributions:** Conceptualization, A.D.; methodology, A.D., J.G. and C.J.; formal analysis, A.D.; investigation, A.D.; writing—original draft preparation, A.D.; writing—review and editing, J.G., C.J. and A.D.; funding acquisition, C.J., A.D. and J.G.; All authors have read and agreed to the published version of the manuscript.

**Funding:** This research was funded by the European Union European Research Development Fund Northern Periphery and Arctic Programme 2014–2020, grant number 304-2559-2016.

**Acknowledgments:** We acknowledge the assistance from WaterPro team members in providing relevant information: Donnacha Doody, Agri-Food and Bioscience Institute, Belfast (for Northern Ireland), Con McLaughlin, C., Donegal County Council, Donegal (for the Republic of Ireland) and Ville Matikka, ELY Centre for Economic Development, Transport and the Environment, Kuopio (for Finland).

**Conflicts of Interest:** The authors declare no conflict of interest.

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
