# Peer review of "An Inventory of Good Management Practices for Nutrient Reduction, Recycling and Recovery from Agricultural Runoff in Europe’s Northern Periphery and Arctic Region"

_water, doi:10.3390/w14132132_

Round 1
Reviewer 1 Report
This manuscript could be suitable for publication in Water Journal, however, the authors are requested to address the following comments while revising the manuscript. The study is within the scope of the journal, however, shows deficiencies and should be significantly improved with the addition of more details about:
- hypothesis in the manuscript
- research questions (RQs) in the end introduction section;
In the method section, please provide the survey research procedure. It is not enough to put information that the data was collected from the project members and presented at the two project meetings held in Iceland. How many questions and to who were addressed in order to obtain information within individual countries e.t.c. Please describe the research methodology step by step.
Conclusions
The conclusion should be presented in line with RQs.
Reviewer 2 Report
The Review Paper, water-1757092: An Inventory of Good Management Practices for Nutrients 2 Reduction, Recycling and Recovery from Agricultural Runoff 3 in Northern Periphery and Arctic (NPA) region
The study consists of a review of the efficiency of good agricultural practices in reducing nutrients, their potential for recycling and recovery of nutrients, the requirements and costs of full operation and maintenance. It also highlights gaps in current knowledge and provides recommendations for future research directions.
This inventory provides comprehensive and a first of a kind guide on available measures and practices to assist regional and local authorities in this region studied, and a good example local communities in many other region.
Is a good manuscript adapt for pubblicvation in this joural, in present form.
Author Response
Thank you very much for your positive comments about our manuscript.
Reviewer 3 Report
The manuscript handles with the good management practices for nutrients reduction, recycling and recovery from agricultural runoff in northern periphery and arctic.
The article presented is an interesting topic and is overall well written and concise. Apparently are not necessarily major modifications.
Author Response

(The authors gave the same response as above.)

Reviewer 4 Report
General Comments:
The review article is designed to study the “An Inventory of Good Management Practices for Nutrients Reduction, Recycling and Recovery from Agricultural Runoff in Northern Periphery and Arctic (NPA) region”. The article cover almost every aspect of the agriculture best management practices for nutrients reduction, recycling and recovery.
The discrepancies of the article are mentioned below to improve its quality.
Revisions:
The article abstract needs revisions. Summarize the each section of the article very well in the article abstract as per journal guideline. See author guidelines.
The introduction section is not well written and need revisions to discuss each aspect of the article in one separate paragraph with more updated citations.
There should be a separate heading about the “Future Research Directions”.
The conclusion of the article also need revision and provide a summary of the article in the conclusion section of the article.
The overall article is good for publication after suggested improvements.
Reviewer 5 Report
The management of pollution, mainly nitrogen and phosphorus coming from agricultural runoff, is of paramount importance in the aspect of water ecosystems protection. Non-point sources of nutrients strongly affect all lakes in agricultural watersheds, posing a threat to water quality, stimulating cyanobacterial blooms and influencing ecosystem services for local community. Therefore, changes in the management practices for croplands are necessary, what is emphasized in many programs e.g. under th EU auspices. It was also underlined in the presented review manuscript, in which an analysis of different GMPs was conducted. Authors have collected broad database of 130 publications, including journal publications and legislatory documents, to select GMPs suitable for Northern Periphery and Arctic areas. All of them were analyzed in the light of their efficiency and nutrient recovery/recycling, to emphasize the gaps in knowledge. In fact, it turned out that many of GMPs efficiency is questionable, what is crucial in the light of funding spend on them.
I find the review very interesting, based on broad literature database and critical view at the data it contain, including gaps recognition. Thus, in my opinion the review shall be published in Water journal.
I only found two repetitions in the text that need correction: first in Introduction (1st paragraph, 8th line - 'and local communities') and in table 4, section 6.1 (NRE - 'highly variable').
Author Response
Point 1: I only found two repetitions in the text that need correction: first in Introduction (1st paragraph, 8th line - 'and local communities') and in table 4, section 6.1 (NRE - 'highly variable').
Response 1: Thank you very much for pointing out these two repetitions, we have corrected them.
Reviewer 6 Report
WATER
An Inventory of Good Management Practices for Nutrients Reduction, Recycling and Recovery from Agricultural Runoff in Northern Periphery and Arctic (NPA) region
Manuscript Number: water-1757092
Article Type: Review Article
General Comment:
Authors reported and research finds that there is a lot of uncertainty, inconsistency, and knowledge gap when it comes to GAPs' performance in terms of nutrient reductions (NRE), the possibility for nutrients recycling and recovery (NRR), operation and maintenance needs (OMR), and costs. Although the impact of GAPs to improved water quality cannot be evaluated, this inventory provides a comprehensive and unique guide to existing measurements and practises to aid regional and municipal governments as well as local populations in the NAP region. The inclusion and retrofitting of Phosphorus Retaining Media is recommended by the authors (PRMs)To fight water pollution, a variety of agricultural management approaches have been developed and widely used as conservation management measures. There has also been broad acknowledgement of the necessity for nitrogen harvesting from wastewaters and resource recovery in the last 10 years. The knowledge in water and runoff management in the Northern Periphery and Arctic (NPA) areas is intermittent and must be increased.
I strongly advise that this paper be published in this journal; I just have a few minor comments, which I have mentioned below.
Specific comments:
- Abstract of this manuscript is well written and perfectly organized.
- Line no.79 to 81. The large number of coupled references are used. It is objectionable. Please specify the particular references like this type of paragraphs.
Despite 5 decades of effort and considerable financial investments in implementa-
tion of GAPs as conservation management strategies for pollution mitigation, excess
loading of nutrients generated by agricultural activities remains a major water quality
issue in Europe and across the globe [3-13].
- Similar objections in the line number 109.
- Source of table 1, should be properly organized for better understanding and looking.
- What do you understand by page in table 1. Its quit confusing.
- Line number 114, references should be specified.
- Table 4. Good Agricultural Practices Nutrients Reduction Efficiencies (NRE), potential for Nutrients Recycling ad Recovery (NRR) and Operation and Maintenance Requirements (OMR) and Costs is very large. Could you please suggest something better plan to make this table.
- Conclusion of this manuscript is missing could you write a fresh conclusion with help of your literature.
- Many old references are used in your introduction section, it should be modified with new ones.
- Rephrase the following paragraph and grammar should be given importance:
“Although the evaluation of GAPs treatment performances and cost-effectiveness has been a subject of investigation for over a decade, both remain largely unknown. For example, the Land and Policy Journal 2010 published 12 scientific papers in a special issue on Soil and Water Conservation Measures in Europe.”
- Ensure that your manuscript is well edited for English language and technical expressions
- All references according to the journal format.
- Include following references for better presentation of your work
Distribution, risk assessment, and source apportionment of polycyclic aromatic hydrocarbons (PAHs) using positive matrix factorization (PMF) in urban soils of East India; Solvent Extraction Coupled with Gas Chromatography for the Analysis of Polycyclic Aromatic Hydrocarbons in Riverine Sediment and Surface Water of Subarnarekha River, Characteristics of Atmospheric Particle-bound Polycyclic Aromatic Compounds over the Himalayan Middle Hills: Implications for Sources and Health Risk Assessment, Runoff water pollution in India, Source apportionment and health risks assessment of black carbon Aerosols in an urban atmosphere in East India
- All old references were kept in your study, I am hereby recommending new references for the better representation of your manuscript
https://doi.org/10.1016/j.scitotenv.2021.151003 https://doi.org/10.1016/j.gsd.2021.100553;
10.1061/(ASCE)HZ.2153-5515.0000586
https://doi.org/10.1007/s10668-020-01167-1
- Your whole manuscript has not formatted. Kindly format it with respect to the journal guidelines
- Many useless full stops, the comma should be deleted and given in the appropriate format.
- Some spelling mistakes are also seen, kindly rectify the same.
End
Round 2
Reviewer 1 Report
Dear Authors,
Type of manuscript: Review
Title: An Inventory of Good Management Practices for Nutrients Reduction, Recycling and Recovery from Agricultural Runoff in Northern Periphery and Arctic (NPA) region - Accept in present form.
Kind regards
Author Response
Dear Reviewer 1,
Thank you very much.
Kind regards,
Aleksandra
Reviewer 6 Report
I am not satisfied with the present Response to Reviewer Comments. I advised the author please read carefully the reviewer’s comments and correct the manuscript according to the reviewer’s advice. The missing reply to the comment is points 6, 8, and 12.
Some comment has not been included.
The author has not added the suggested references in spite of this he added unreverent and old references.
